# KCNN4 Promotes the Stemness Potentials of Liver Cancer Stem Cells by Enhancing Glucose Metabolism

**DOI:** 10.3390/ijms23136958

**Published:** 2022-06-23

**Authors:** Jing Fan, Ruofei Tian, Xiangmin Yang, Hao Wang, Ying Shi, Xinyu Fan, Jiajia Zhang, Yatong Chen, Kun Zhang, Zhinan Chen, Ling Li

**Affiliations:** 1Department of Cell Biology, National Translational Science Center for Molecular Medicine, Fourth Military Medical University, Xi’an 710005, China; fan6688jing@163.com (J.F.); tianruofei@fmmu.edu.cn (R.T.); yxiangmind@163.com (X.Y.); yingshi0703@163.com (Y.S.); xinyulion@163.com (X.F.); zjj12405681160326@163.com (J.Z.); chenamy@fmmu.edu.cn (Y.C.); zkdyx-1@163.com (K.Z.); 2Department of Cell Biology, Institutes of Biomedicine, Jinan University, Guangzhou 510632, China; puzhika3720@163.com

**Keywords:** LCSCs, *KCNN4*, metabolism plasticity, glycolysis, OXPHOS

## Abstract

The presence of liver cancer stem cells (LCSCs) is one of the reasons for the treatment failure of hepatocellular carcinoma (HCC). For LCSCs, one of their prominent features is metabolism plasticity, which depends on transporters and ion channels to exchange metabolites and ions. The K^+^ channel protein *KCNN4* (Potassium Calcium-Activated Channel Subfamily N Member 4) has been reported to promote cell metabolism and malignant progression of HCCs, but its influence on LCSC stemness has remained unclear. Here, we demonstrated that *KCNN4* was highly expressed in L-CSCs by RT-PCR and Western blot. Then, we illustrated that *KCNN4* promoted the stemness of HC-C cells by CD133^+^CD44^+^ LCSC subpopulation ratio analysis, in vitro stemness transcription factor detection, and sphere formation assay, as well as in vivo orthotopic liver tumor formation and limiting dilution tumorigenesis assays. We also showed that *KCNN4* enhanced the glucose metabolism in LCSCs by metabolic enzyme detections and seahorse analysis, and the *KCNN4*-promoted increase in LCSC ratios was abolished by glycolysis inhibitor 2-DG or OXPHOS inhibitor oligomycin. Collectively, our results suggested that *KCNN4* promoted LCSC stemness via enhancing glucose metabolism, and that *KCNN4* would be a potential molecular target for eliminating LCSCs in HCC.

## 1. Introduction

Hepatocellular carcinoma (HCC) is one of the most intractable human malignancies and the fifth leading cause of cancer-related deaths worldwide [1,2,3]. The biological mechanism of hepatocarcinogenesis and its progression are complex. Accumulating evidence indicates the significance of liver cancer stem cells (LCSCs), which are small-population liver cancer cells with the potential for self-renewal and tumorigenicity, in tumorigenesis, recurrence, and metastasis [4,5]. Despite numerous studies having revealed the molecular basis for LCSCs, which includes genetic mutations, epigenetic disruption, signaling pathway dysregulation, and/or microenvironment remodeling [5], none of them have brought about an effective solution for eradicating LCSCs.

In recent years, the contribution of energy metabolism to CSCs has attracted increasing attention in the field [6,7,8]. Strategies targeting CSC metabolic characteristics are regarded as new opportunities for cancer treatments. LCSCs are featured by both addicted glycolysis and sustained oxidative phosphorylation (OXPHOS), which means that glucose flows more to the glycolysis pathway, [9,10,11], while fatty acid and glutamate contribute to the enhanced OXPHOS [12,13]. Inhibiting glycolysis or fatty acid oxidation (FAO) by targeting specific regulatory molecules or key metabolic enzymes has been used to attenuate stem cell potentials [14,15]. For example, glycolytic inhibitor 2-deoxy-D-glucose (2-DG) has drastically inhibited the tumorigenicity and sorafenib resistance of LCSCs [16]. However, the plasticity of its metabolic phenotype is the main obstacle in targeting LCSCs [4,5]. Therefore, the energy sources for LSCSs, other than the metabolic phenotypes, need to be better understood.

The absorption and utilization of metabolic substrates depend on the transporters or channels mediating the exchanges of essential nutrients and/or ions between the intracellular compartment and surrounding microenvironment [17,18]. In most cases, nutrients and ions are transported together, for example, Na^+^-driven glucose symport. These facts imply that the alteration in channel protein activities may influence the processes of nutrient utilization and the associated metabolism plasticity. 

Herein, in this paper, we have screened a channel protein KCa3.1 that has a significantly high level in LCSC. KCa3.1, encoded by Potassium Calcium-Activated Channel Subfamily N Member 4 (*KCNN4*) [19], has been reported to be highly expressed in various cancers including HCC [20,21,22,23,24,25,26,27], affecting cell metabolism, and reducing cell motility in glioblastoma derived CSCs [28,29]. However, its involvement in LCSCs is still unknown. Therefore, we investigated the expression level of *KCNN4* in LCSCs and explored the role of *KCNN4* in promoting LCSC stemness. Our results may help to provide a novel molecular biomarker and therapeutic target for LCSCs. 

## 2. Results

### 2.1. KCNN4 Was Highly Expressed in Hepatocellular Carcinoma Stem Cells (LCSCs)

The mRNA stemness index (mRNAsi), an index that describes the similarity between cancer cells and stem cells, is considered a CSC semi-quantitative form that integrates the signatures of mRNA expression for stemness phenotype from the HCC patient information in the TCGA database. Higher mRNAsi scores are connected with higher CSC potentials and greater tumor dedifferentiation [30]. Therefore, we used mRNAsi to explore the possible genes involved in LCSC characteristics, for example, oncogene or epigenetic modifier mutations, and mRNA/miRNA transcriptional networks and signaling pathways. We divided RNA-seq data of 377 cases from human LIHC (Liver hepatocellular carcinoma) samples into a low-differentiated state group and a high-differentiated state group. Out of 32,191 genes, 2463 up-regulated genes and 221 down-regulated genes were screened (Figure 1A; Appendix A). Then, Gene ontology (GO) and the Kyoto Encyclopedia of Genes and Genomes (KEGG) pathway enrichment analysis for differential genes were performed (Figure 1B,C; Appendix A). 

In the above KEGG pathway enrichment analysis, apart from PI3K-Akt, cAMP, and the ECM-receptor signal pathway that have been extensively studied in stem cells, the protein digestion and absorption pathway that listed as the secondary significantly altered pathwayand functionally associated with cell metabolism attracted our attention. As the contribution of metabolism to stemness potentials was one of the most vital characteristics of LCSCs, we tried to find the key molecules that influence LCSCs metabolism. Then, the interaction network of genes within this pathway was analyzed in the STRING database (https://string-db.org/ (accessed on 2 October 2021)). The key genes were distributed into five parts: digestive enzymes (e.g., *CPA1*), extracellular matrix (e.g., *COL4A4*), solute carrier family (e.g., *SLC1A1*), ATPase Na+/K+ transporting (*ATP1A2*, *ATP1A3*, and *ATP1A4*), and potassium channel family (*KCNN4*, *KCNQ1*, and *KCNJ13*) (Figure 1D). Interestingly, the cation homeostasis and ion channel activity associated pathways, such as calcium ion homeostasis and potassium ion transmembrane transporter activity, also occurred in the GO and KEGG enrichment analysis (Figure 1B,C; Appendix A). Therefore, we focused on the relationship between potassium channel protein and LCSCs. 

Among the enriched key genes in the potassium channel family, *KCNN4,* which codes KCa3.1 protein, is activated by intracellular calcium, then mediates potassium efflux and calcium influx; the latter has been reported to influence LCSC stemness [31]. Based on the analysis above, we proposed that *KCNN4* may be associated with LCSC stemness. To validate this, we detected the expression of *KCNN4* in LCSCs, which enriched as tumorspheres from two HCC cell lines: Huh7 and HepG2. As shown in Figure 1E,F and Appendix A, the expression of *KCNN4* was higher in LCSCs than in non-CSCs both at the mRNA and protein levels. Furthermore, the expression level of *KCNN4* was validated in the CD44^−^ and CD44^+^ subpopulations (Appendix A). In the following, we proceeded to explore the connection of *KCNN4* with LCSC stemness.

### 2.2. KCNN4 Promoted In Vitro Stem Cell Potentials of LCSCs

To determine the potential of *KCNN4* in LCSCs, we effectively knocked down the expression of *KCNN4* in HepG2 and Huh7 cell lines using siRNA (si-858) (Appendix A) and shRNA lentivirus (sh-*KCNN4*-1 and/or sh-*KCNN4*-2) (Appendix A), respectively. Flow cytometry showed that the ratios of the CD133^+^CD44^+^ subpopulation were significantly declined both in HepG2 cells (3.49 ± 0.62% vs. 7.74 ± 1.75% for si-*KCNN4*, 3.84 ± 0.17% vs. 7.40 ± 0.14% for sh-*KCNN4-2,* respectively) and in Huh7 cells (1.47 ± 0.87% vs. 5.36 ± 0.71% for si-*KCNN4*, 2.10 ± 0.4% vs. 4.43 ± 0.73% for sh-*KCNN4-2*, Figure 2A,C). QPCR and Western blot showed that the expression of stem cell transcription factors *SOX2*, *OCT4*, and *NANOG* were all significantly decreased after *KCNN4* knockdown (Figure 2B,D below; Appendix A). Sphere formation analysis showed a significant decrease in sphere numbers both in HepG2 cells (35 ± 16 vs. 54 ± 7 for sh-*KCNN4*-1, 21 ± 8 vs. 54 ± 7 for sh-*KCNN4*-2, respectively) and in Huh7 cells (27 ± 6 vs. 46 ± 7 for sh-*KCNN4*-1, 22 ± 5 vs. 46 ± 7 for sh-*KCNN4*-2) (Figure 2E). 

Furthermore, we applied gain-of-function strategy by constructing *KCNN4*-overexpressed cells (Appendix A). As expected, with the overexpression of *KCNN4*, the ratios of the CD133^+^CD44^+^ subpopulation (9.11 ± 0.1 vs. 7.49 ± 0.52% for HepG2, 5.29 ± 0.54% vs. 3.65 ± 0.35% for Huh7, Figure 2C), expression level of stem cell transcription factor *SOX2*, *OCT4*, and *NANOG* (Figure 2D, above; Appendix A), and the numbers of spheres (88 ± 31 vs. 58 ± 15 for HepG2, 69 ± 14 vs. 49 ± 2 for Huh7, Figure 2E) were all significantly increased.The resistance to radiotherapy and chemotherapy is one of the main characteristics of CSCs. Therefore, we treated cells with different concentrations of sorafenib and gemcitabine for 48 h, and detected cell growth and apoptosis. Generally, the survival rate of both groups was decreased with the increase in drug concentration (Appendix A). However, under the same drug concentration, the survival rate of the *KCNN4* overexpressing group was significantly higher than that of the ctrl group and the sh-*KCNN4* group. In addition, we observed a higher apoptosis rate in the sh-*KCNN4* and enhanced drug resistance in ov-*KCNN4*, as compared to control cells (Appendix A). These results suggested that *KCNN4* promoted the in vitro stem cell potentials of LCSCs, thus indicating the roles of *KCNN4* in promoting tumor development.

### 2.3. KCNN4 Promoted Tumor Formation by Enhancing In Vivo Stem Cell Potentials of LCSCs

To explore the differences in tumorigenicity of the ov-*KCNN4*/sh-*KCNN4* cell lines and stemness potential of CSCs in vivo, we established two different mouse models. Firstly, in an orthotopic xenograft model of nude mouse, *KCNN4*-knockdown resulted in a reduction in tumor incidence (1/7 vs. 5/7) (Figure 3A, Appendix A) and loss of liver weight (1.24 ± 0.06 vs. 1.46 ± 0.09 g, Figure 3B). Conversely, *KCNN4*-overexpression promoted tumorigenicity and tumor growth, which was indicated by a significantly increased tumor incidence (7/7 vs. 5/7) (Figure 3A, Appendix A) and liver weight (1.88 ± 0.14 vs. 1.46 ± 0.09 g, Figure 3B). These data revealed that *KCNN4* was important for the tumorigenicity of HCC.

Then, in the gradient dilution model of NOD-SCID mouse, to further explore the role of *KCNN4* in tumor initiation, CSCs enriched from tumorspheres of lv-*ctrl*, ov-*KCNN4,* and sh-*KCNN4*-2 were inoculated in each mouse individually at different numbers: 2 × 10^2^, 2 × 10^3^, 2 × 10^4^, and 2 × 10^5^. Similar with nude mouse, ov-*KCNN4* bearing mice displayed significantly larger tumor volume and higher tumor weight while sh-*KCNN4* mice showed significantly smaller volume and lower weight, as compared with the lv-*ctrl* group in each level (Figure 3C–E; Appendix A). As compared with the lv-*ctrl* group, median tumor incidence times were significantly longer in ov-*KCNN4* but shorter in sh-*KCNN4* mice (Figure 3E; Appendix A). In mice inoculated with 2 × 10^5^ cells, for example, the median tumor incidence times were 25 ± 3, 43 ± 1, and 32 ± 1 days for ov-*KCNN4*, sh-*KCNN4,* and lv-*ctrl* cells, respectively. Furthermore, we analyzed the tumor-initiating ability in mice bearing xenografts with different levels of *KCNN4*. We found that tumor formation efficiency of 100% (9/9) occurred in mice injected with 2 × 10^3^, 2 × 10^4^, and 2 × 10^5^ lv-*ctrl* CSCs, which occurred in all four cell numbers of ov-*KCNN4* CSC-bearing mice but only in 2 × 10^4^ or 2 × 10^5^ sh-*KCNN4* CSC-inoculated mice. More importantly, the expression levels of *KCNN4* were not only vital to the tumor-initiating abilities but also to the stemness potentials with the LCSC frequency, which were 1/1, 1/1629, and 1/182 for ov-*KCNN4*, sh-*KCNN4,* and lv-*ctrl* CSCs, respectively (Appendix A). Altogether, these results suggested that *KCNN4* promoted tumor formation by enhancing the ratio of stem cell population in HCC.

### 2.4. KCNN4 Enhanced Metabolic Fitness in LCSCs by Upregulating Glucose Metabolism

CSCs are featured by stemness and metabolic plasticity [15]. Thus, we measured ECAR (Extracellular Acidification Rate) and OCR (Oxygen Consumption Rate) in both non-stem HCC cells and LCSCs. As expected, LCSCs showed significantly higher ECAR and higher OCR as compared to non-stem HCC cells (Figure 4A,B), which was reflected by an increase in basal respiration, proton leak, ATP production, and glycolysis reserve in both HepG2 and Huh7 cells (Appendix A). Next, we detected the mRNA levels of rate-limiting enzymes in glycolysis, OXPHOS, and mitochondrial biogenesis. Compared with non-CSCs, LCSCs expressed higher levels of rate-limiting enzymes in the glycolysis pathway, including *LADH*, *GLUT1*, *HK2,* and *PFK1* in both HepG2 and Huh7 cells (Figure 4C,D). However, highly expressed OXPHOS (*PDHA*, *PDHB*, and *PDHK*) and mitochondrial biosynthesis-related genes (*TFAM*, *PGC1a*) only occurred in HepG2s, in contrast to lowly expressed OXPHOS, but highly expressed mitochondrial biosynthesis-related genes in Huh7 cells (Figure 4E,F). Combining together, these results suggested that LCSCs exhibited enhanced glycolysis and mitochondrial biogenesis compared with non-CSCs.

We further explored whether the expression of *KCNN4* affected LCSC metabolism. Firstly, the gene sets of HCC patients were grouped according to the expression level *KCNN4*, and the expression of key metabolic enzymes was analyzed. The expression of glucose metabolism-associated enzymes ALDH1L2, HK1, HK3, G6PD, and mitochondrial enzymes PKM and PDHA was higher in the *KCNN4*-High groups than in the *KCNN4*-Low groups. Positive correlation with *KCNN4* (Appendix A) and GSEA showed that glycolysis and gluconeogenesis, fatty acid metabolism, and glutamate metabolism pathways were significantly enriched in *KCNN4*-High (Appendix A). We also found that both non-CSCs and CSCs in *KCNN4* overexpressed cells showed significantly higher extracellular acidification rates and higher oxygen consumption rate/maximal respiration/spare respiratory capacity (Figure 5A–D; Appendix A). For the expression of rate-limiting enzymes, the mRNA levels of *GLUT1*, *PKM, PDHK,* and *TFAM* were significantly changed according to the alteration of *KCNN4* in both cells, while only the mRNA levels of *HK2* and *PFK1* changed accordingly in HepG2 (Figure 5E–G). These results suggested that *KCNN4* promoted the activation of the glycolysis pathway as well as enhanced the oxidative respiration and biogenesis of mitochondria in HCC cells including LCSCs, which reflected that the role of *KCNN4* in promoting stemness potentials was connected to enhancing cell metabolism. 

### 2.5. KCNN4-Enhanced Glucose Metabolism Promoted Stem Cell Potentials of LCSCs

To investigate whether *KCNN4* promoted the stem cell potentials by enhancing glucose metabolism, we treated lv-*KCNN4* cells with 2-DG (2-Deoxy-D-arabino-hexose, a glucose analog that inhibits glycolysis) or oligomycin (an inhibitor of H^+^-ATP-synthase that blocks oxidative phosphorylation and electron transport chain), respectively. As shown in Figure 6A, the increased ratios of the CD133^+^CD44^+^ subpopulation in HepG2 by *KCNN4* overexpression were significantly abolished by 2-DG (2.83 ± 0.16% to 1.68 ± 0.08%, *p* < 0.0001) or oligomycin (2.83 ± 0.16% to 2.42 ± 0.12%, *p* = 0.0080). In Huh7 cell lines, similar phenomena were found (3.80 ± 0.44% to 2.18 ± 0.15% for 2-DG, *p* = 0.0002; 3.80 ± 0.44% to 2.85 ± 0.11%, *p* = 0.0056 for oligomycin, respectively). Consistently, compared to HepG2 lv-*KCNN4*, the ratio of ALDH^+^ cells was significantly inversed from 63.93 ± 8.34% to 24.07 ± 6.49% as treated with 2-DG (*p* = 0.0002), and from 63.93 ± 8.34% to 44.50 ± 1.85% as treated with oligomycin (*p* = 0.0163) (Figure 6B). In Huh7, similar trends (50.30 ± 2.71% to 21.90 ± 5.14% for 2-DG, *p* < 0.0001, and to 38.57 ± 1.63% for oligomycin, *p* = 0.0061) were also observed. Moreover, in comparison, 2-DG had a greater inhibition rate than oligomycin did (Figure 6C). These results indicated that KCNN4 enhances the stemness potentials of LCSCs by increasing glucose metabolism 

## 3. Discussion

In this study, we found that *KCNN4* was highly expressed in LCSCs, where *KCNN4* promoted the ratio of the CD133^+^CD44^+^ subpopulation, the expression of stem cells transcription factors, and the sphere formation ability in vitro, as well as increased tumor incidence and tumor growth in vivo. Moreover, we showed that *KCNN4* facilitated glucose metabolism of LCSCs, that is, both glycolysis and OXPHOS. Finally, we demonstrated that the enhanced glucose metabolism participated in the enhanced stem potency caused by *KCNN4*. These data suggested that *KCNN4* plays a novel oncogenic role in HCC tumor development and might become a novel treatment target for drug-resistant LCSCs.

In the present studies, abnormal expression and/or function of KCa3.1 has been reported to be involved in cancer, autoimmune disorder, and vascular inflammation [32]. Similar to our findings, abnormal expression of KCa3.1 was found in both cholangiocarcinoma and HCC [28,29,33,34,35], promoting cell proliferation [28], and invasion and metastasis [29]. In our study, we have found for the first time that KCa3.1 is highly expressed in LCSCs and has an important role in promoting stemness potentials and tumor initiation in HCC. However, it was worth reflecting that we observed a lower expression of *KCNN4* in the SI-HIGH; although, the KCNN4 was positively correlated with stem cell transcription factors in TCGA datasets (Appendix A), which were supposed to be higher. Considering that the expression level of *KCNN4* was generally lower in liver cells while higher in immune cells (Appendix A) (http://biogps.org/#goto=welcome (accessed on 28 April 2022)), and it has been reported that tumors with SI-HIGH stemness subtypes had rare immune infiltration [36], it may be that the levels of *KCNN4* in immune cells overwhelm the expression value on the whole. In fact, the immune-infiltrated analysis showed the correlations between the expression of *KCNN4* and immune infiltration of 22 immune cell types in patients with HCC (Appendix A). These results showed that *KCNN4* may play an important role in the tumor microenvironment, both in tumor cells and immune cells. 

Moreover, we have demonstrated that the potassium channel protein KCa3.1 enhanced the tumor-initiating ability of LCSCs, which may be achieved by enhancing the glucose metabolism of LCSCs. Although it is unknown whether the KCNN4-mediated metabolism promotion of LCSCs was dependent on its function of transporting ions, our data here indicate that channels associated with nutrient utilization may be the reason for increasing the LSCSs’ stemness potential. Mechanically, ion channels are one of the communication pathways for nutrient exchange between tumor cells and the tumor microenvironment [37,38]. Material transport and signal exchange across the ion channel in the plasma membrane are the basis of the interaction. More molecular targets remain to be discovered, which is the basis for developing novel treatments. The role of *KCNN4* in regulating oxygen consumption and ATP production has been confirmed in a subset of Pancreatic carcinoma cell lines [39], but metabolic regulation of *KCNN4* in the progression of other cancers has not been reported. Combined targeting KCNN4 with its potential downstream signaling molecules, such as Glut1, may provide new possibilities for addressing tumor drug resistance. The mechanism of regulation of glucose metabolism needs to be explored in the future.

In the present studies, as the key bioenergetic organelles, mitochondria harbor several proteins with proven or hypothetical ion channel functions. These channels make an important contribution to the regulation of mitochondrial function, such as energy transducing processes, reactive oxygen species production, and mitochondrial integrity [40]. KCa3.1 is expressed in the plasma membrane and mitochondrial intima, encoded by the same gene, *KCNN4*. In HCT116 cells, mtKCa3.1 channels are located in the inner mitochondrial membrane and regulated by the mitochondrial matrix Ca^2+^ concentration [41]. Given that K+-flux is involved in maintaining the mitochondrial function, which drives the respiratory chain, regulates mitochondrial volume, and as a consequence regulates mitochondrial respiratory function [42], mtKCa3.1 must be necessary for the maintenance of mitochondrial function. The regulation of mtKCa3.1 on mitochondrial oxidative phosphorylation has preliminarily been observed [39], but more evidence in different human tumors remains to be verified. In-depth mechanisms of KCa3.1-mediated metabolism and functional implications should be elucidated in the future, both in the plasma membrane and mitochondrial intima. In our study, we preliminarily explored the effect of KCa3.1 on cellular metabolic phenotypes at the global level but cannot distinguish the specific effects of channels between KCa3.1 in the plasma membrane and mtKCa3.1 on some particular metabolic pathway; this would be the next step for us to explore.

In conclusion, our study suggests that KCa3.1 can promote stemness of tumor cells by enhancing glucose metabolism. Our results suggest that the ion channel may be a novel target for the cell metabolism of LCSCs. Based on these findings, treatment of liver cancer may be beneficial with potassium channel inhibitors. However, the regulation mechanism of KCa3.1 on biological behavior of stem cells needs to be further explored. The elucidation of membrane-targeted and mitochondrial-subtargeted potassium channels on the effects of stem cell biological behaviors including metabolism characteristics would be helpful to further explain the potentials of stem cells. 

## 4. Materials and Methods

### 4.1. Database and mRNAsi Analysis

The RNA sequencing (RNA-Seq) data and FPKM data from 366 samples of human LIHC (liver hepatocellular carcinoma) samples were downloaded from the TCGA database (the Cancer Genome Atlas, http://tcga-data.nci.nih.gov/tcga/ (accessed on 5 December 2019)). These data were current as of 5 December 2019. We obtained SI of LIHC patient samples. For the LIHC dataset, the repeated genes were removed firstly, and the genes with a sum equal to 0 were removed. All the tumor samples were divided into high-SI and low-SI groups by quantile level of SI [30], and divided into high-KCNN4 and low-KCNN4 groups by quantile gene expression level of KCNN4. 

### 4.2. Differentially Expressed Genes

Differential gene analysis was performed by using R package edgeR for RNA-seq count data [43]. Identification of differentially expressed genes (DEGs) were selected using the following criteria: (a) fold change > 2; (b) *p* < 0.05. Using the limma and pheatmap packages to draw volcano plots and heatmaps in R.

### 4.3. Pathway Enrichment Analysis and PPI Analysis

Gene ontology (GO) and Kyoto Encyclopedia of Genes and Genomes (KEGG) analyses were performed using R package Clusterprofiler [44]. A *p*-value < 0.05 and an FDR < 0.05 were considered statistically significant. Protein–protein interaction networks (PPI) were performed by using STRING database (https://string-db.org/ (accessed on 2 October 2021)).

### 4.4. Gene set Enrichment Analysis (GSEA)

GSEA was performed using the “clusterProfiler” [44] and “GSEABase” R packages to find enriched terms in metabolic pathways. Differences were considered statistically significant if *p* < 0.05 or FDR (false discovery rate) q < 0.25 and |NES| > 1.

### 4.5. Evaluation of Immune Cell Infiltration 

The infiltration of 22 immune cell types in the AMI and IS samples used in this study were analyzed using CIBERSORT [45]. The expression data were imported into CIBERSORT using R and then iterated 1000 times to estimate the relative proportion of each immune cell type.

### 4.6. Cell Culture and Establishment of Stable Cell Sublines

The human liver cancer cell lines Huh7 and HepG2 were originally obtained from the China Center for Type Culture Collection (CCTCC) and stored in National Translational science Center for Molecular Medicine with mycoplasma and short tandem repeat (STR) profiling tested (Cenvino, Beijing, China). Huh7 and HepG2 were cultured in DMEM (Thermo, Waltham, MA, USA) supplemented with 10% fetal bovine serum (FBS, HAKATA), 5% Penicillin–Streptomycin and 5% L-glutamine in an atmosphere containing 5% CO_2_ at 37 °C. The tumor spheres were cultured with DMEM/F12 serum-free media and performed more than three passages, these cells were sought to be CSCs. All cells were passaged every 2 to 3 days. The GV208 and *KCNN4* lentiviral shRNA constructs were obtained from Genechem (Shanghai, China), which were inserted into a luciferase tag. *KCNN4* overexpressed construct was established by subcloning human *KCNN4* cDNA into the GV208 lentiviral expression vector. The lentivirus carrying *KCNN4* shRNA or CDNA and their individual corresponding control vector were injected into liver cancer cells, and the stable subclones were selected by adding 2 μg/mL puromycin and verified by analyzing *KCNN4* expression levels using qPCR and Western blot. RELN small interfering RNA (siRNA) was used for transient transfection. The sequence of shRNA and siRNA as well as their negative control are shown in Appendix A. 

### 4.7. Obtaining of LCSCs 

LCSCs were obtained using two methods. Tumorspheres were enriched from culture medium (Appendix A). The digested cells were resuspended with sphere medium and were inoculated in ultra-low attachment 6-well plates at the density of 10,000 cells/3 mL/well. Then, cells were incubated in a CO_2_ incubator at 37 °C and were passaged more than three times. In FACs sorting, the HepG2 and Huh7 cells were adjusted to 1 × 10^7^ cells/mL with serum-free medium, were added to an appropriate amount of direct labeled flow antibody FITC-CD44 (Biolegend, San Diego, CA, USA), and were incubated for 30 min at 4 °C. Labeled cells were washed with serum-free medium twice and samples resuspended to sort with BD Aria III.

### 4.8. Quantitative Real-Time PCR

RNA was extracted using a Total RNA Kit II (OMEGA Bio-tek, Norcross, GA, USA) under RNase-free conditions and was reversely transcribed into cDNA using a PrimeScript^TM^ RT reagent kit (TaKaRa, San Jose, CA, USA). Single-stranded cDNA was amplified using TB Green premix Ex Taq II (TaKaRa) on a QuantStudio 7 Flex Real-Time PCR System (Applied Biosystems, Waltham, MA, USA). The quantified transcripts from the samples were normalized to the expression of house-keep gene β-actin, and calculated using the 2^−^^ΔΔCT^ method. The primers involved are shown in Supplement Appendix A. 

### 4.9. Western Blot

Cells were harvested, lysed, and quantified by BCA protein assay kit (Beyotime Biotechnology, Shanghai, China). Equal amounts of total protein lysates were resolved by 12% SDS-PAGE for 2 h, and then were transferred to PVDF membranes (Millipore, Bedford, MA, USA) for 1 h. Then, 5% skimmed milk solution was applied to block non-specific binging for 2 h. Afterward, the membranes were incubated in primary antibodies: anti-KCa3.1 (Abbiotec, 1:200), anti-tubulin (Abcam, 1:3000), anti-Sox2 (Abcam, 1:800), anti-Oct4 (Abcam, 1:800), and anti-Nanog (Abcam, 1:1000) at 4 °C overnight. Anti-rabbit secondary antibody (Abcam, 1:3000) and anti-mouse secondary antibody (Abcam, 1:5000) were used to detect the primary antibody. Finally, imaging was carried out with a digital imaging system to detect the bands relating to the proteins of interest.

### 4.10. Tumor Sphere Formation Assays

Culture medium for sphere culture was prepared according to the formula shown in Appendix A. The digested cells were resuspended with sphere medium and were inoculated in an ultra-low attachment 96-well plate at a density of 50 cells/200 μL/well. Cells were then incubated in a CO_2_ incubator at 37 °C for 1–2 weeks and were passaged once every 3–5 days. Finally, spheres larger than 50 μm in diameter were counted at high magnification. 

### 4.11. Analysis of CD133^+^CD44^+^ Subpopulation

The cells were digested and suspended in serum-free medium, and the cellular concentration was adjusted to 1 × 10^7^ cells/mL. To the cell suspension was added the appropriate amount of direct labeled flow antibody FITC-CD133 (Biolegend), PE-CD44 (Biolegend), and it was incubated for 30 min at 4 °C. Labeled cells were washed with serum-free medium twice and samples resuspended with 300–500 μL/tube serum-free medium and were detected with BD LSRFortessa. Data analysis was performed in Flowjo.6.1. 

### 4.12. Analysis of ALDH^+^ Subpopulation

The ALDEFLUOR kit (Stem Cell Technologies, Vancouver, BC, Canada) was applied to analyze the ratio of ALDH^+^ subpopulation; 1 × 10^6^ cells were incubated in the ALDEFLUOR assay buffer containing 1 μM ALDH1 substrate BAAA at 37 °C for 40–60 min, whereas negative control cells were incubated with 5 μL of ALDH inhibitor diethylaminobenz aldehyde (DEAB) under the same conditions. Labeled cells were washed twice and samples resuspended with 300 μL ALDHEFLUOR assay buffer and were detected with BD LSRFortessa. Data analysis was performed in Flowjo.6.1.

### 4.13. CCK8 Assay

Cell proliferation was detected using the CCK8 assay (cell counting kit-8, beyotine). Briefly, HepG2 and Huh7 cells (3 × 10^3^ cells/well) were seeded in 96-well plates and treated with drugs. The concentrations for sorafenib were 0, 2.5, 5, 15/0, 1.25, 2.5, and 7.5 umol/L, and the concentrations for gemcitabine were 0, 2.5, 10, 20/0, 1.25, 5, and 10 umol/L in HepG2/Huh7. PBS was used as the drug control. After incubating the cells with drugs for 24 h and 48 h, 100 μL CCK8 solution was added to each well and it was incubated at 37 °C for 2 h before the absorbance was measured at 450 nm. 

### 4.14. Apoptosis Analysis

Apoptosis was examined by Annexin V-7-AAD Apoptosis Detection Kit. HepG2 and Huh7 cells (2 × 10^5^ cells/well) were seeded in 6-well plates and treated with sorafenib (15 μmol in HepG2 and 7.5 μmol in Huh7). After 24 h, cells were suspended with 100 μL Annexin V Binding Buffer including 2.5 μL Annexin V-APC and 2.5 μL 7-AAD staining solution, and incubated for 15–20 min at room temperature away from light. Then, 400 μL Annexin V Binding Buffer was added and cells were detected with BD LSRFortessa. Data analysis was performed by Flowjo.6.1.

### 4.15. Orthotopic Xenograft Model of Nude Mouse 

For the in situ tumorigenesis experiment, three groups of cells (HepG2 ov-*KCNN4*, HepG2 lv-*ctrl*, and HepG2 sh-*KCNN4*), which, at logarithmic growth stage, were digested and resuspended with serum-free medium to the concentration of 10^7^ cells/mL. SPF male Balb/C-nude mice at 4–5 weeks old were anesthetized by 1% pentobarbital sodium. Then, the abdomen of the mice was opened up, and the above-prepared cells were injected under the liver capsule in the left lobe of the liver with insulin needle (50 μL/mice), then the wound was sutured and disinfected with iodine. Afterward, nude mice were fed with normal diet and observed. Then, 45 d after injection, in vivo fluorescence imaging was performed. In detail, mice were anesthetized with isoflurane and then intraperitoneally injected with 3 mg substrate Luciferase (Beyotime Biotechnology); 10 min later, fluorescence imaging was performed. Data analysis was performed in Creamstyle, and the mouse cervical vertebrae were severed, and the liver with tumor was cheeked and photographed. 

### 4.16. NOD-SCID Mice Subcutaneous Gradient Tumorigenesis Model

For subcutaneous gradient tumorigenesis, CSCs enriched from tumorspheres of three groups cells (HepG2 ov-*KCNN4*, HepG2 lv-*ctrl*, and HepG2 sh-*KCNN4*) were digested and resuspended with PBS: Matrigel =1:1 for different concertation. Male NOD-SCID mice at 4–5 weeks old (9 mice/group) were divided into three groups according to the above three CSCs, and the back of each mouse was subcutaneously inoculated with 2 × 10^2^, 2 × 10^3^, 2 × 10^4^, and 2 × 10^5^ same cells, respectively. The NOD-SCID mice were fed with normal diet. The time of tumor formation and tumor growth at each site were observed and recorded, and tumor formation curves were drawn. After 75 days, the tumor grew to the observation end point, and the mice were killed by neck removal. The tumor masses at each subcutaneous site were weighed and photographed, and tumor volume was calculated according to V = length × width^2^/2. Stem cell frequency was analyzed according to the extreme limiting dilution analysis (ELDA), as previous study has shown [46].

### 4.17. Seahorse—Mito-Stress Test

Oxygen consumption rate (OCR) was measured using the Seahorse XFe96 Analyzer (Agilent Technologies, Santa Clara, CA, USA). Briefly, adherent cells were inoculated in appropriate numbers (10,000 cells/well) in pre-hydrated 96-well plates and incubated at 37 °C with 5% CO_2_ for 24 h. The next day, suspension cells from passaged tumor microspheres were plated into the precoated wells and the adherent cells were equilibrated with pre-warmed Seahorse XF assay medium (DMEM base medium with 10 mM glucose, 2 mM L-glutamine, and 1 mM sodium pyruvate, pH 7.4), then cultured cell plates at 37 °C without CO_2_ for 1 h. OCR of the cellular monolayer was measured before (basal level) and after the sequential injection of 1 μM oligomycin, 0.5 μM FCCP, and 0.5 μM Rot/antimycin A. The experiment consists of at least five biological repeated samples. Data were normalized to the cell numbers of each well and analyzed in WAVE.

### 4.18. Seahorse—Glycolysis Stress Test 

Extracellular acidification rate (ECAR) was measured using the Seahorse XFe96 Analyzer (Agilent Technologies) with similar procedures to OCR analysis except for different treatments. In detail, ECAR was measured before (basal level) and after the sequential injection of 10 mM glucose, 1 μM oligomycin, and 50 mM 2-DG. 

### 4.19. Statistical Analysis

All experiments were performed independently at least three times, and all results were presented as the mean ± SD. Data were analyzed using GraphPad prism 8.0 (San Diego, CA, USA), The statistical significance of the difference between two separate experimental groups was determined by a two-tailed unpaired student’s *t*-test, and the difference between four separate experimental groups was determined by one-way ANOVA. *p* < 0.05 was considered as statistically significant.

## Figures and Tables

**Figure 1 ijms-23-06958-f001:**
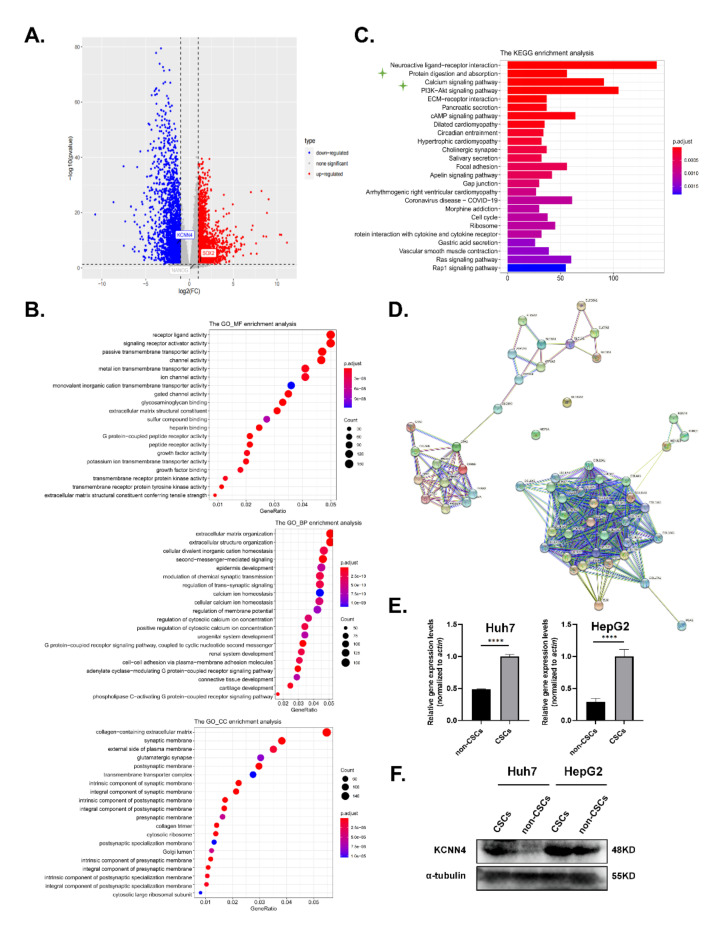
*KCNN4* was highly expressed in hepatocellular carcinoma stem cells: (**A**,**C**) The RNA sequencing (RNA-Seq) expression data of 377 cases of human LIHC samples were downloaded from the TCGA database (http://tcga-data.nci.nih.gov/tcga/ (accessed on 5 December 2019)), and we divided the sample into two groups with low-differentiated state and high-differentiated state according to the stemness index for the following analysis. As shown in the Volcano Plot (**A**), 2463 up-regulated genes and 2214 down-regulated genes were screened out of 32,191 genes. GO (**B**) and KEGG (**C**) pathway enrichment analysis for differential genes were visible. (**D**) The interaction network of genes in enriched pathways in the STRING database (https://string-db.org/ (accessed on 2 October 2021)) was analyzed to search for known protein interactions. (**E**) Relative mRNA expression of *KCNN4* in non-CSCs and CSCs enriched from tumorspheres. (**F**) Protein expression of *KCNN4* in non-CSCs and CSCs enriched from tumorspheres. These data represent the mean ± SD from at least three independent experiments. Two-way Student’s *t*-test. **** *p* < 0.0001.

**Figure 2 ijms-23-06958-f002:**
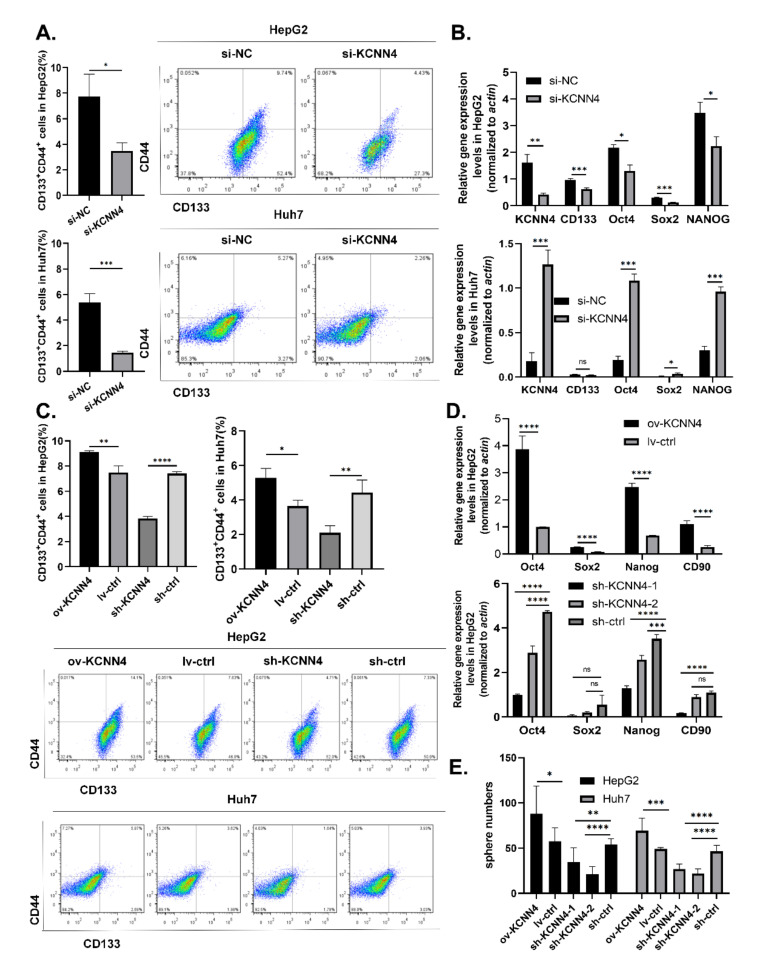
***KCNN4* enhanced stemness of two HCC cell lines in vitro:** (**A**) CD133^+^CD44^+^ CSC subpopulations were detected by flow cytometry in HepG2 and Huh7. (**B**) The expressions of stem cell transcription factors were examined by qPCR after *KCNN4* knockdown by siRNA in HepG2 and Huh7. (**C**) CD133^+^CD44^+^ CSC subpopulations were detected by flow cytometry in *KCNN4* overexpression and knockdown in HepG2 and Huh7. (**D**) The expression of stem cell transcription factors was examined by qPCR after *KCNN4* overexpression or knockdown in HepG2. (**E**) Tumorsphere formation assay in HepG2 and Huh7 cell lines. Histogram shows the number of cell spheres over 50 μm in diameter (magnification = 4×, scale = 50 μm). These data represent the mean ± SD from at least three independent experiments. Two-way Student’s *t*-test. * *p* < 0.05, ** *p* < 0.01, *** *p* < 0.001, **** *p* < 0.0001.

**Figure 3 ijms-23-06958-f003:**
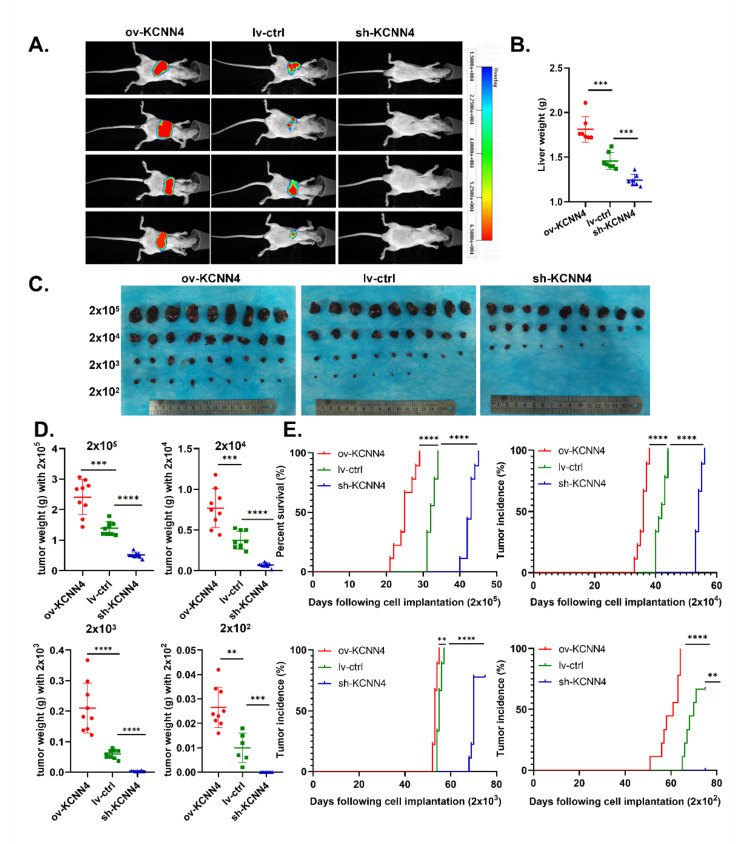
***KCNN4* promoted tumor formation and stem potency in vivo:** (**A**) In situ model of hepatocellular carcinoma was established in nude mice. The HCC cells were labeled with luciferase. The images were taken of mouse xenograft tumor at 45 days. The liver of tumor-bearing mice was observed and the tumor formation was observed and weighed (**B**). (**C**) In the gradient dilution model of the NOD-SCID mouse, tumor-bearing mice were divided into four groups injected with different CSCs: 2 × 10^2^, 2 × 10^3^, 2 × 10^4^, and 2 × 10^5^. The mice were killed at 75 days after tumor bearing and the tumors were dissected and photographed. (**D**,**E**). In the gradient dilution model of the NOD-SCID mouse, tumor weight (g), and tumor incidence time (days) were shown. These data represent the mean ± SD from at least three independent experiments. Each dot represented a mouse. Two-way Student’s *t*-test. ** *p* < 0.01, *** *p* < 0.001, **** *p* < 0.0001.

**Figure 4 ijms-23-06958-f004:**
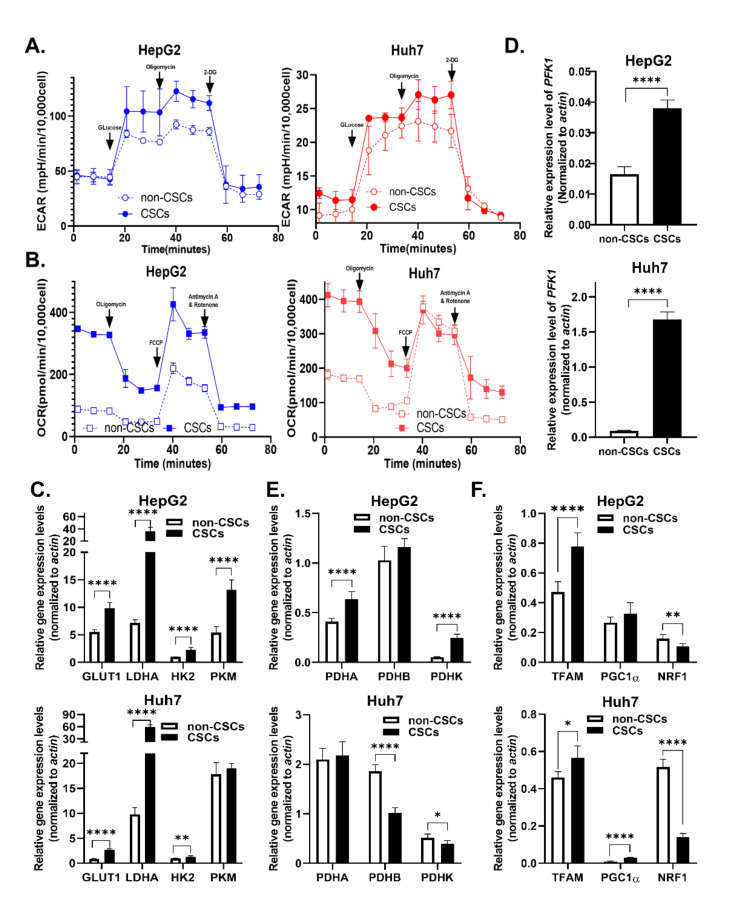
**Glucose metabolism was elevated in LCSCs compared to non-stem cells:** (**A**,**B**) OCR and ECAR in CSCs and non-CSCs were measured in two cell lines. (**A**) Glycolysis-stress test, and (**B**) Mito-stress test. Relative expression of rate-limiting enzyme genes of metabolic pathways in CSCs and non-CSCs was detected by qPCR and is shown as follows. (**C**,**D**) mRNA expression of glycolysis pathway key enzymes, including *LADH, GLUT1, HK2, PFK1,* and *PKM*. (**E**) mRNA expression of pyruvate dehydrogenase complex (*PDHC*), which includes *PDHA, PDHB,* and *PDHK. PDHC* plays an important role in mitochondrial oxidative respiration chain energy metabolism. (**F**) mRNA expression of mitochondrial biosynthesis key enzymes, including *TFAM*, *PGC1a*, and *NRF1*. These data represent the mean ± SD from at least three independent experiments. Two-way Student’s *t*-test. mRNA expression levels were normalized by β-actin endogenous control. * *p* < 0.05, ** *p* < 0.01, **** *p* < 0.0001.

**Figure 5 ijms-23-06958-f005:**
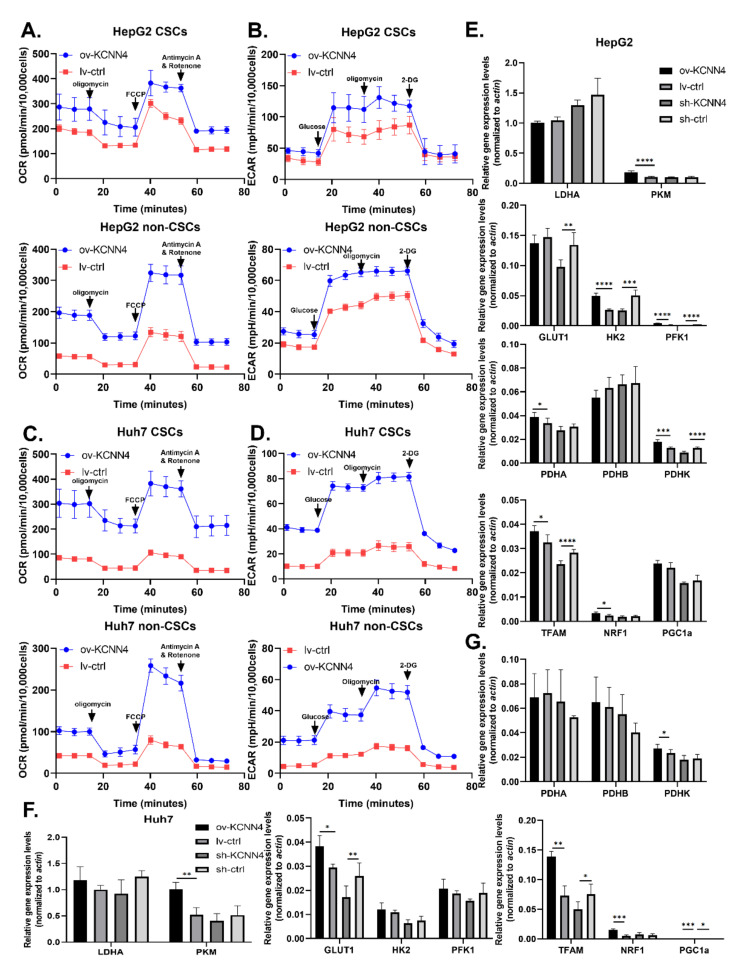
***KCNN4* enhances metabolic fitness in LCSCs by upregulating glucose metabolism:** (**A**–**D**) OCR and ECAR were measured in non-CSCs and CSCs, respectively. Mito-stress test was shown as (**A**,**C**), and glycolysis stress test as (**B**,**D**). Relative expression of rate-limiting enzyme genes of metabolic pathways in CSCs and non-CSCs of two cell lines, in HepG2 (**E**) and in Huh7 (**F**,**G**). mRNA expression of glycolysis pathway key enzymes, including *LADH, GLUT1, HK2, PFK1,* and *PKM*, pyruvate dehydrogenase complex (*PDHC*), and mitochondrial biosynthesis key enzymes, including *TFAM, PGC1a,* and *NRF1,* were detected as (**E**–**G**). These data represent the mean ± SD from at least three independent experiments. Two-way Student’s *t*-test. mRNA expression levels were normalized by *β-actin* endogenous control. * *p* < 0.05, ** *p* < 0.01, *** *p* < 0.001, **** *p* < 0.0001.

**Figure 6 ijms-23-06958-f006:**
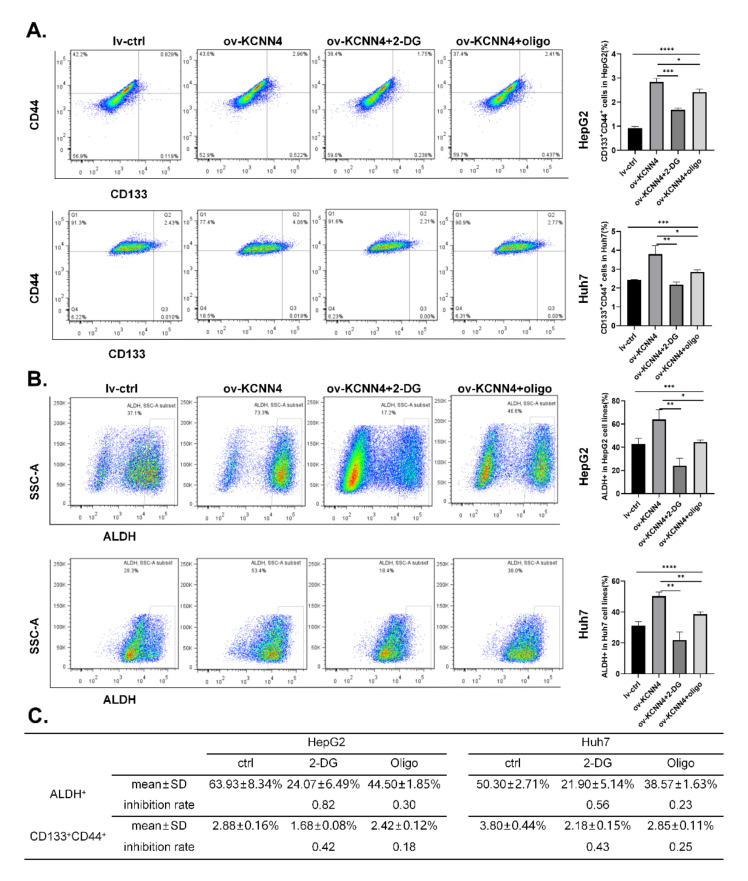
**Enhanced glucose metabolism militated enhanced stem cell potency:** lv-*KCNN4* cells were treated with 2-DG and oligomycin respectively, and the ratio of CD133^+^CD44^+^ cells (**A**) or ALDH^+^ cells (**B**) was detected by flow cytometry. (**C**) The inhibition rate of 2-DG and oligomycin in Huh7 and HepG2. These data represent the mean ± SD from three independent experiments. One-way ANOVA was used to detect differences between groups, and 2-way Student’s *t*-test was used to detect differences between two groups. * *p* < 0.05, ** *p* < 0.01, *** *p* < 0.001, **** *p* < 0.0001.

## Data Availability

In this study, the RNA sequencing (RNA-Seq) expression data of 377 cases of human LIHC samples were downloaded from the TCGA database (http://tcga-data.nci.nih.gov/tcga/ (accessed on 5 December 2019)). These data were current as of 5 December 2019.

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
