# Peer review of "KCNN4 Promotes the Stemness Potentials of Liver Cancer Stem Cells by Enhancing Glucose Metabolism"

_ijms, 2022, doi:10.3390/ijms23136958_

Round 1

Reviewer 1 Report

The paper by Dr. Jing Fan and collaborators is intended to demonstrate the role of KCNN4 in promoting stemness abilities in liver cancer cells by boosting glucose metabolism.

KCNN4 participation in cancer has been already known for several years. In liver cancer its involvement has been focused mainly on cell invasion and metastasis with no significant information on potential role of KCNN4 on cancer stem cell subpopulation. In that sense this study would contribute to the general knowledge of cancer stem cell abilities and potentially identify new therapeutic avenues.

There are several points of concern in the study, mainly in the in vitro and in vivo experiments:

  1. Authors distinguish in vitro CSC from non-CSC by cell ability to form spheres under non-attachment culture plates. So, in those experiments in which they analyze KCNN4 gene expression in CSC versus non-CSC, where do non-CSC cells come from? Do they use 2D cell cultures?
  2. In glucose metabolism experiments, it needs also to be defined how CSC and non-CSC cell populations are obtained.
  3. Is KCNN4 increasing cell proliferation per se, independently of cell stemness potential? A set of in vitro experiments determining proliferative index in ov-KCNN4 cells versus non-transfected KCNN4 cells is mandatory. This is crucial in all experiments that are done. For instance, in glucose metabolism experiments. If KCNN4 is increasing cell proliferation rate, glucose metabolism will be increased also. It is also important in animal experiments in which a higher cell proliferation rate would explain the results obtained in tumor formation in vivo.
  4. If authors consider CD133+CD44+ as true LCSCs population, tumor formation in vivo experiments should have been designed to compare CD133+CD44+ tumorigenicity potential against CD133-CD44- subpopulation. This way, all cells are theoretically ov-KCNN4 and a KCNN4 unwanted side effect could be ruled-out.
  5. Paper of Malta et al (reference #30) defines higher stemness index in liver cancer samples from TCGA, by high expression of CYCLINB1, ACC1 and low expression PD-L1 and ANNEXIN-A1. So, an increase in stemness capabilities should be mirrored by an increase in CYCLINB1, ACC1?
  6. Supplementary Table 3, L-glutamine units say 2mm/ml surely there is a typo error.

Author Response

RE: Resubmission of our manuscript (Manuscript ID: ijms-1729822)

Title: KCNN4 Promotes the Stemness Potentials of Liver Cancer Stem Cells by Enhancing Glucose Metabolism

Dear reviewer: 

Thank you very much for your email with encouraging news regarding our manuscript. We also thank the reviewers for their positive/constructive comments and suggestions, which truly helped us to improve our manuscript. After incorporating their comments, I would like to re-submit our revised manuscript for your consideration to be published in the section of “Molecular Pathology, Diagnostics, and therapeutics >>Tumor Microenvironment and its Actor: Are ion Channels Relevant?” in the International Journal of Molecular Sciences. The revisions are highlighted in red in the revised manuscript, and our point-by-point answers to the reviewers’ comments are attached below. Besides, we have carefully checked the whole manuscript and corrected some format, spelling and grammatical errors, as well as some errors in the figures, as required in the author’s guidelines.

Thank you again, and I hope that the revision is acceptable.

Best regards,

Our responses to the reviewers’ comments:

  1. Authors distinguish in vitro CSC from non-CSC by cell ability to form spheres under non-attachment culture plates. So, in those experiments in which they analyze KCNN4 gene expression in CSC versus non-CSC, where do non-CSC cells come from? Do they use 2D cell cultures?

We really appreciate all your comments and suggestions. On this issue, it is true that we utilized the different methods for obtaining in vitro CSCs and non-CSCs. We initially choose the passaged tumorspheres as the source of LCSC, which is also a commonly used strategy for in vitro enriching functional CSCs and the adherent cells as the source of non-CSCs. For CSCs, we utilized ultra-low attachment plates and cultured them under serum-free suspension conditions as described in Supplementary Table 3 and then passaged cells for more than three generations. For non-CSCs, we utilized a serum-containing medium and performed adherent 2D conditions. In addition, according to Reviewer 2’s suggestion, we sorted CD44- and CD44+ subpopulations by flow cytometry as non-CSCs and CSCs, respectively.

  1. In glucose metabolism experiments, it needs also to be defined how CSC and non-CSC cell populations are obtained.

We fully agree with the reviewer and appreciated your suggestion. We added related descriptions in the Methods section “cells culture and establishment of stable cell sublines” and “seahorse” of the revised manuscript accordingly.

  1. Is KCNN4 increasing cell proliferation per se, independently of cell stemness potential? A set of in vitro experiments determining proliferative index in ov-KCNN4 cells versus non-transfected KCNN4 cells is mandatory. This is crucial in all experiments that are done. For instance, in glucose metabolism experiments. If KCNN4 is increasing cell proliferation rate, glucose metabolism will be increased also. It is also important in animal experiments in which a higher cell proliferation rate would explain the results obtained in tumor formation in vivo.

  We fully agree with your constructive comment. According to the cancer stem cell theory, CSCs are a small proportion of cancer cells with the potential for self-renewal and tumorigenicity. Usually, CSCs can be activated and entered into proliferate status upon tumor environmental stimuli by performing symmetric division (Metabolism and the Control of Cell Fate Decisions and Stem Cell Renewal. Annu Rev Cell Dev Biol. 2016 Oct 6;32:399-409). In the latter proliferated situation, stem cells can be expanded and their number or ratio is increased, and then CSCs play an important role during the persistent growth of tumors. Quiescent CSCs and activated state CSCs are in constant transition. Furthermore, tumor cells are highly plastic, and cells in the non-stem status could change into stem status.

Based on the analysis above, we assumed that the roles of KCNN4 increasing cell proliferation of CSCs may dependent on stemness potentials, in other words, the roles of KCNN4 on glucose metabolism and in vivo tumor formation are mainly derived from its influence on stemness potential, although its influences on cell proliferation cannot be excluded. As the previous study has already shown that KCNN4 promotes cell proliferation in two HCC cell lines that including Huh7 (CCK8 assay in Figure 2 c-f, EdU staining assay in Figure 2 k,l; The potassium channel KCa3.1 promotes cell proliferation by activating SKP2 and metastasis through the EMT pathway in hepatocellular carcinoma. Int J Cancer. 2019 Jul 15;145(2):503-516.), we did not detect proliferative index in KCNN4 overexpressed or knock-down cells in our manuscript.

In our glucose metabolism experiments, to exclude the influences of cell proliferation, we tested cell viability after culturing cells for 12 hours, which period is the exact culture time before OCR or ECAR were detected. The CCK-8 assay results are shown in the following (CCK-8 assay in 12h), which revealed that there were no significant differences in cell viability among ov-KCNN4, ctrl and sh-KCNN4 groups. Therefore, the influence of cell proliferation on glucose metabolism could be excluded.

Finally, to explore the influence of KCNN4 expression on tumorigenicity and stemness potential of CSCs in vivo, we established two different mouse models. In the orthotopic xenograft model of nude mice, we injected mice with ov-KCNN4, lv-ctrl, and sh-KCNN4 cells; while in the gradient dilution model of the NOD-SCID mice, we injected mice with CSCs enriched from tumorspheres. The latter showed that the expression levels of KCNN4 are not only vital to the tumor-initiating ability but also to the stemness potential, which can be reflected by the LCSCs frequency (1/1, 1/1629 and 1/182 for ov-KCNN4, sh-KCNN4 and lv-ctrl cells respectively, Supplementary Table2). We are so sorry for the confusion caused to you by our unclear description and we have added the related description in the Results section in red (Figure 3).

Figure 2 c- f, CCK8 assay; Figure 2 k, l. EdU staining assay.

(Figures were cited from Int J Cancer. 2019 Jul 15;145(2):503-516.)

CCK8 assay;

(Absorbance in 450nm of ov-KCNN4, ctrl and sh-KCNN4 after incubating 12hours)

  1. If authors consider CD133+CD44+ as a true LCSCs population, tumor formation in vivo experiments should have been designed to compare CD133+CD44+ tumorigenicity potential against CD133-CD44- subpopulation. This way, all cells are theoretically ov-KCNN4 and a KCNN4 unwanted side effect could be ruled out.

We are very sorry about the confusion caused by the unclear description. Actually, in the orthotopic xenograft model of nude mice, we have injected into the liver with ov-KCNN4, lv-ctrl, and sh-KCNN4 cells; however, in the gradient dilution model of the NOD-SCID mice, we have subcutaneously injected CSCs that enriched by passaged tumorspheres from ov-KCNN4, lv-ctrl, and sh-KCNN4 cells. For more clarity, we have added the detailed description in the Results section for Figure 3 in red.

As cell surface biomarkers especially CD133 are plastic, its expression level varies greatly under different conditions, and the ratio of CD133+CD44+ LCSC subpopulations being as low as approximately 5% in HepG2 and 2% in Huh7, we adopted passaged tumorspheres as the source of CSCs in our manuscript, which is also a commonly used strategy for in vitro enriching functional CSCs. However, to make the experimental data more convinced, the tumorigenicity potential of CD133+CD44+ CSCs should also be tested. In the future, we would add these data as you suggested.

  1. Paper of Malta et al (reference #30) defines higher stemness index in liver cancer samples from TCGA, by high expression of CYCLINB1, ACC1 and low expression PD-L1 and ANNEXIN-A1. So, an increase in stemness capabilities should be mirrored by an increase in CYCLINB1, ACC1?

We are extremely grateful to reviewer for addressing us about the defining new metrics of cancer stemness by stemness indexes in the paper of Malta et al (reference #30). In that paper, the author provides stemness indices for assessing the degree of oncogenic dedifferentiation, thus higher stemness index is assumed to be connected with higher stemness capabilities. As dedifferentiated cells can be arisen from the inherent long-lived stem or via dedifferentiation from non-stem cancer cells that are induced by the tumor microenvironment, and as stemness indexes have a high degree of intratumor heterogeneity which can be affected by stromal cells, hypoxia, and infiltration of immune cells, therefore, an increase in stemness capabilities not necessary be mirrored by an increase in higher stemness index genes like CYCLINB1 and ACC1. For example, they observed negative associations between stemness and EMT gene signatures. As they stated in the article that “the relationship between EMT and stemness remains a hotly debated topic, with several studies showing that EMT is necessarily associated with stemness”. Taking together, specifically answering this question still requires more laboratory experiments for validation. The limitations caused by sample sites, the heterogeneity of sequencing tissues, the influence of data analysis methods, and other unknown factors, all of which make the prediction of related genes have certain limitations. Future works should focus more on verifying the significance of these higher stemness index-related genes in stemness capabilities.

6/Supplementary Table 3, L-glutamine units say 2mm/ml surely there is a typo error.

We are very sorry for the typo error about the L-glutamine units. L-glutamine units should be 2mm/l but not 2mm/ml. We revised it accordingly in Supplementary Table 3.

Reviewer 2 Report

In this manuscript, authors show that the potassium channel KCNN4 is expressed in liver cancer cells enriched in stem cells (LCSCs), and that KCNN4 knockdown decreases the CD133+CD44+ cell subpopulation, the expression of stemness transcription factors, the sphere formation, as well as the liver tumor size after in vivo orthotopic HCC cells injection suggesting that KCNN4 contributes to the stemness of HCC cells. They also show that KCNN4 overexpression enhances the glucose metabolism in LCSCs. Their results are convincing and suggest that KCNN4 could promote LCSC stemness via enhancing glucose metabolism.

However, this manuscript would be even more convincing if authors could complete their study with the following experiments:

1/ Authors showed that KCNN4 is overexpressed in LCC cells cultured as spheres that they claim to be LCSC enriched cells. Showing that KCNN4 is overexpressed in CD44+/CD133+ cells FACS sorted and amplified would strengthen their claim that KCNN4 is overexpressed in LCSC and promotes stemness.

2/ Authors showed a decreased expression of the stem cell transcription factors SOX2, OCT4 and Nanog in KCNN4 knockdown cells at the RNA level. A western-blot showing that these stem cell transcription factors are decreased at the protein level would strengthen these results and the link between KCNN4 and stemness.

3/ Authors suggest that inhibiting KCNN4 could improve liver cancer treatment. The comparison of the efficacy of chemotherapy such as Gemcitabine or Doxorubicin on control HCC cells, cells overexpressing KCNN4 and KCNN4 knockdown cells could strengthen this hypothesis.

Other comments:

  • Define FAO (line 48)
  • Increase the quality of Figure 1 especially of Fig 1D
  • Fig 2D: in Huh7 KCNN4 knockdown cells, KCNN4, OCT4, SOX2 and Nanog expression is increased instead of being decreased as expected and as observed in HepG2 KCNN4 knockdown cells. Please, correct the figure.

Author Response

RE: Resubmission of our manuscript (Manuscript ID: ijms-1729822)

Title: KCNN4 Promotes the Stemness Potentials of Liver Cancer Stem Cells by Enhancing Glucose Metabolism

Dear reviewer: 

Thank you very much for your email with encouraging news regarding our manuscript. We also thank the reviewers for their positive/constructive comments and suggestions, which truly helped us to improve our manuscript. After incorporating their comments, I would like to re-submit our revised manuscript for your consideration to be published in the section of “Molecular Pathology, Diagnostics, and therapeutics >>Tumor Microenvironment and its Actor: Are ion Channels Relevant?” in the International Journal of Molecular Sciences. The revisions are highlighted in red in the revised manuscript, and our point-by-point answers to the reviewers’ comments are attached below. Besides, we have carefully checked the whole manuscript and corrected some format, spelling and grammatical errors, as well as some errors in the figures, as required in the author’s guidelines.

Thank you again, and I hope that the revision is acceptable.

Best regards,

Our responses to the reviewers’ comments:

  1. Authors showed that KCNN4 is overexpressed in LCC cells cultured as spheres that they claim to be LCSC enriched cells. Showing that KCNN4 is overexpressed in CD44+/CD133+ cells FACS sorted and amplified would strengthen their claim that KCNN4 is overexpressed in LCSC and promotes stemness.

We thank the reviewer for your constructive comments. We initially choose the passaged tumorsphere as the source of LCSC based on the following considerations.

 First, CSC is a functional definition of subpopulation that has self-renewal potential, which usually can be enriched from tumorsphere by culturing cells in suspension serum-free medium in vitro at a low cell number. To exclude aggregates formed under suspension serum-free conditions, the tumorspheres that passaged more than three generations are usually considered as CSCs.

Secondly, cell surface biomarkers especially CD133 are plastic and closely related to culture environment, cell status, and artificial operation to cells. Their expression levels vary greatly under different conditions internally and externally. Thirdly, the ratio of CD133+CD44+ LCSC subpopulations is as low as approximately 5% in HepG2 and 2% in Huh7. Obtaining enough cells for qPCR and western blot analysis is usually technically challenging. Followed by the reviewer’s insightful suggestion, we obtained CSCs and non-CSCs by sorting CD44- and CD44+ subpopulations instead. Consistently, the expression level of KCNN4 is higher in CD44+ subpopulations than in CD44- subpopulations for both cell lines. We have added the corresponding description in the Methods and Result section, and also added the data in the new Supplementary figure S1D.

  1. Authors showed a decreased expression of the stem cell transcription factors SOX2, OCT4 and Nanog in KCNN4 knockdown cells at the RNA level. A western-blot showing that these stem cell transcription factors are decreased at the protein level would strengthen these results and the link between KCNN4 and stemness.

Following the reviewer’s suggestion, we have examined the protein expression of stem cell-related transcription factors, SOX2, OCT4, and Nanog. We obtained similar results as with the mRNA expression of SOX2, OCT4, and Nanog. These results are shown in the new Supplementary FigureS2E.

  1. Authors suggest that inhibiting KCNN4 could improve liver cancer treatment. The comparison of the efficacy of chemotherapy such as Gemcitabine or Doxorubicin on control HCC cells, cells overexpressing KCNN4 and KCNN4 knockdown cells could strengthen this hypothesis.

Followed by the reviewer’s constructive suggestion, the role of KCNN4 in drug resistance was explored by CCK8 assay and apoptosis analysis. We treated KCNN4 overexpressed or knockdown cells with different concentrations of sorafenib and gemcitabine for 24 and 48 hours and then detected cell proliferation and apoptosis ratios. Our results showed that the survival or apoptotic ratio under sorafenib and gemcitabine treatment was increased or decreased accordingly as KCNN4 overexpressed, while were decreased or increased accordingly as the KCNN4 knockdown. These results indicated that the efficacy of chemotherapy (gemcitabine) or target therapy (sorafenib) in eradicating HCC cells could be enhanced by inhibiting the expression of KCNN4. These new data are added in Supplementary figure S3. And, we have added the corresponding description in the methods and result section.

Other comments:

  1. Define FAO (line 48)

We thank the reviewer’s suggestion. We have annotated FAO (fatty acid oxidation) in the manuscript in red.

  1. Increase the quality of Figure 1, especially of Fig 1D

We thank the reviewer’s suggestion. We have increased the quality of Figure 1, especially of Figure 1B, 1D by inserting original high-resolution images in the manuscript. And, we have packaged the original images in the supplementary materials for reviewing.

  1. Fig 2D: in Huh7 KCNN4 knockdown cells, KCNN4, OCT4, SOX2 and Nanog expression is increased instead of being decreased as expected and as observed in HepG2 KCNN4 knockdown cells. Please, correct the figure.

We apologize for the mislabeling in Figure 2D (below panel), in which the title of Y-axis should be “Relative gene expression levels on HepG2” but was mistyped by “Relative gene expression levels on Huh7”. We have revised the labeling in the revised manuscript in red.

Reviewer 3 Report

In the research paper entitled “KCNN4 Promotes the Stemness Potentials of Liver Cancer Stem Cells by Enhancing Glucose Metabolism”, the authors tried to show the role of KCNN4 in promoting stemness in liver cancer stem cells. The paper is quite interesting and the experimental approaches, the authors used to address the question are impressive. There are a few minor points to be noted:

1. The authors should highlight a few important Stem cell marker genes in the volcano plot of Fig1A along with KCNN4.

2. Please adjust the figure labeling in the Figure2

Author Response

RE: Resubmission of our manuscript (Manuscript ID: ijms-1729822)

Title: KCNN4 Promotes the Stemness Potentials of Liver Cancer Stem Cells by Enhancing Glucose Metabolism

Dear reviewer:

Thank you very much for your email with encouraging news regarding our manuscript. We also thank the reviewers for their positive/constructive comments and suggestions, which truly helped us to improve our manuscript. After incorporating their comments, I would like to re-submit our revised manuscript for your consideration to be published in the section of “Molecular Pathology, Diagnostics, and therapeutics >>Tumor Microenvironment and its Actor: Are ion Channels Relevant?” in the International Journal of Molecular Sciences. The revisions are highlighted in red in the revised manuscript, and our point-by-point answers to the reviewers’ comments are attached below. Besides, we have carefully checked the whole manuscript and corrected some format, spelling and grammatical errors, as well as some errors in the figures, as required in the author’s guidelines.

Thank you again, and I hope that the revision is acceptable.

Best regards,

Our responses to the reviewers’ comments:

  1. The authors should highlight a few important Stem cell marker genes in the volcano plot of Fig1A along with KCNN4.

Followed by the reviewer’s constructive suggestion, we have highlighted the gene’s name for KCNN4 and stem cell marker SOX2, Nanog in Figure1A. Consistent with Supplementary figure S5C, we observed a lower expression of KCNN4 in the SI-HIGH group that was supposed to be higher, although the KCNN4 was positively correlated with stem cell transcription factors in TCGA datasets (Supplementary figure S5D). Considering that the expression level of KCNN4 was generally lower in liver cells while higher in immune cells (Supplementary figure S6), and that tumor tissues with SI-HIGH stemness subtypes had rare immune infiltration, for example, in T cells CD8, Macrophages M0, NK cells resting, T cells CD4 memory activated (Supplementary figure S5E), it may be that the levels of KCNN4 in immune cells overwhelm the expression value on the whole level. We addressed this related explanation in the Discussion sections accordingly.

  1. Please adjust the figure labeling in the Figure2

We thank the reviewer. We have revised the figure labeling in the revised manuscript.

Round 2

Reviewer 2 Report

The authors responded to my comments and improved their manuscript accordingly. The manuscript is now acceptable for publication in IJMS in my opinion.